# Estimation of COVID-19 spread curves integrating global data and borrowing information

**Se Yoon Lee***, **Bowen Lei**, **Bani Mallick**

Department of Statistics, Texas A&M University, College Station, Texas, United States of America

* seyoonlee@stat.tamu.edu

**Data Availability Statement:** The data of cumulative number of COVID-19 infected cases are available from COVID-19 Data Repository by the Center for Systems Science and Engineering (CSSE) at Johns Hopkins University (github.com/

## Abstract

Currently, novel coronavirus disease 2019 (COVID-19) is a big threat to global health. The rapid spread of the virus has created pandemic, and countries all over the world are struggling with a surge in COVID-19 infected cases. There are no drugs or other therapeutics approved by the US Food and Drug Administration to prevent or treat COVID-19: information on the disease is very limited and scattered even if it exists. This motivates the use of data integration, combining data from diverse sources and eliciting useful information with a unified view of them. In this paper, we propose a Bayesian hierarchical model that integrates global data for real-time prediction of infection trajectory for multiple countries. Because the proposed model takes advantage of borrowing information across multiple countries, it outperforms an existing individual country-based model. As fully Bayesian way has been adopted, the model provides a powerful predictive tool endowed with uncertainty quantification. Additionally, a joint variable selection technique has been integrated into the proposed modeling scheme, which aimed to identify possible country-level risk factors for severe disease due to COVID-19.

## Introduction

Since Thursday, March 26, 2020, the US leads the world in terms of the cumulative number of infected cases for a novel coronavirus, COVID-19. On this day, a dashboard provided by the Center for Systems Science and Engineering (CSSE) at the Johns Hopkins University (https://systems.jhu.edu/-) [1] reported that the numbers of the confirmed, death, and recovered from the virus in the US are 83,836, 1,209, and 681, respectively. Fig 1 displays daily infection trajectories describing the cumulative numbers of infected cases for eight countries (US, Russia, UK, Brazil, Germany, China, India, and South Korea), spanning from January 22nd to May 14th, which accounts for 114 days. The dotted vertical lines on the panel mark certain historical dates that will be explained. As seen from the panel, the US has been a late-runner until March 11th in terms of the infected cases, but the growth rate of the cases had suddenly skyrocketed since the day, and eventually excelled the forerunner, China, just in two weeks, on March 26th. Fig 2 shows the cumulative infected cases for 40 countries on May 14th: on the day, the

CSSEGISandData/COVID-19). The data sources for the covariates can be downloaded from the World Bank Data (data.worldbank.org), World Health Organization Data (apps.who.int/gho/data/node. main), and National Oceanic and Atmospheric Administration (noaa.gov). In supporting Information, we also provided a minimal underlying data set and relevant R codes.

**Funding:** Research reported in this publication was supported by National Cancer Institute of the National Institutes of Health under award number R01CA194391, and NSF grant no. NSF CCF-1934904. The funders had no role in study design, data collection and analysis, decision to publish, or preparation of the manuscript.

**Competing interests:** The authors have declared that no competing interests exist.

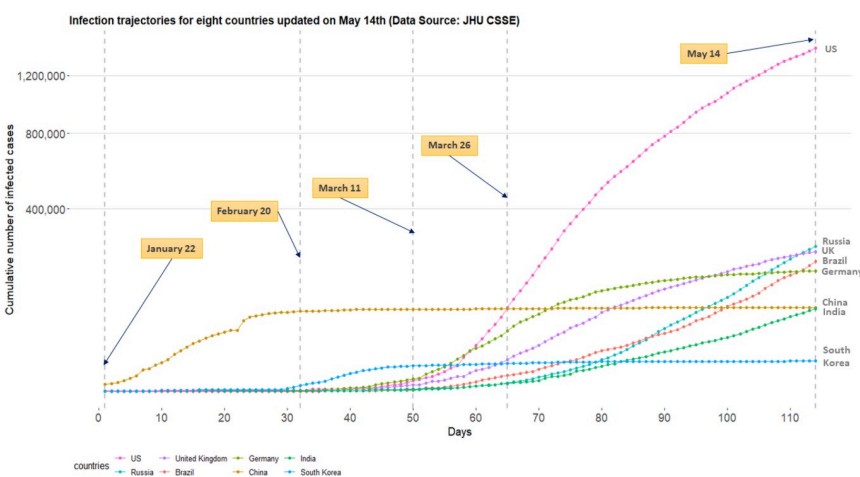

**Fig 1. Daily trajectories for cumulative numbers of COVID-19 infections for eight countries (US, Russia, UK, Brazil, Germany, China, India, and South Korea) from January 22nd to May 14th.** (Data source: Johns Hopkins University CSSE).

number of cumulative infected cases for the US was 1,417,774 which is more than five times of that of Russia, 252,245.

Since the COVID-19 outbreak, there have been numerous research works to better understand the pandemic in different aspects [2–9]. Some of the recent works from statistics community are as follows. [2] focused on a serial interval (the time between successive cases in a chain of transmissions) and used the gamma distribution to study the transmission on Diamond Princess cruise ship. [3] proposed the generalized susceptible exposed infectious removed model to predict the inflection point for the growth curve, while [4] modified the proposed model and considered the public health interventions in predicting the trend of COVID-19 in China. [5] proposed a differential equation prediction model to identify the influence of public policies on the number of patients. [8] used a symmetrical function and a long tail asymmetric function to analyze the daily infections and deaths in Hubei and other places in China. [6] used an exponential model to study the number of infected patients and

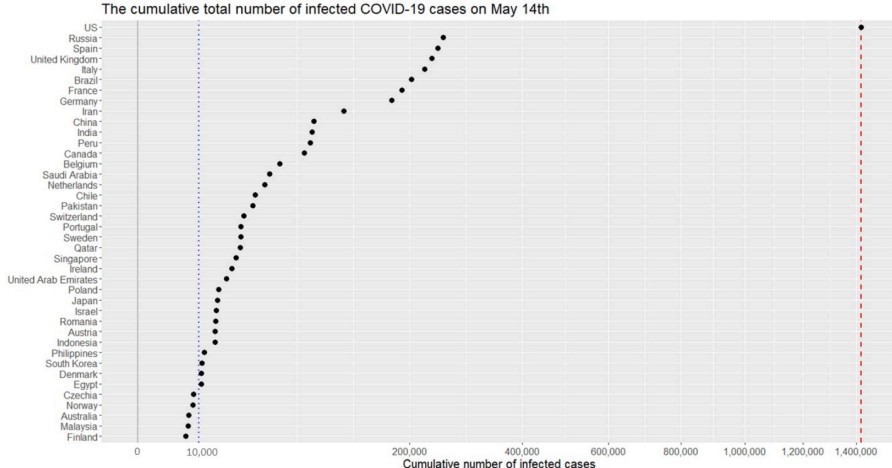

**Fig 2. Cumulative numbers of infected cases for 40 countries on May 14th.** (*x*-axis are scaled with squared root for visualization purpose.) The red dashed vertical lines represents 1,417,774 cases.

patients who need intensive care in Italy. One of the major limitations of these works is that the researches are confined by analyzing data from a single country, thereby neglecting the global nature of the pandemic.

One of the major challenges in estimating or predicting an infection trajectory is the heterogeneity of the country populations. It is known that there are four stages of a pandemic: visit economictimes.indiatimes.com/-. The first stage of the pandemic contains data from people with travel history to an already affected country. In stage two, we start to see data from local transmission, people who have brought the virus into the country transmit it to other people. In the third stage, the source of the infection is untraceable. In stage four the spread is practically uncontrollable. In most of the current literature, estimation or prediction of the infection trajectory is based on a single country data where the status of the country falls into one of these four stages. Hence, such estimation or prediction may fail to capture some crucial changes in the shape of the infection trajectory due to a lack of knowledge about the other stages. This motivates the use of data integration [10, 11] which combines data from different countries and elicits a solution with a unified view of them. This will be particularly useful in the current context of the COVID-19 outbreak.

Recently, there are serious discussions all over the world to answer the crucial question: "even though the current pandemic takes place globally due to the same virus, why infection trajectories of different countries are so diverse?" For example, as seen from Fig 1, the US, Italy, and Spain have accumulated infected cases within a short period of time, while China took a much longer time since the onset of the COVID-19 pandemic, leading to different shapes of infection trajectories. It will be interesting to find a common structure in these infection trajectories for multiple countries, and to see how these trajectories are changing around this common structure. Finally, it is significant to identify the major countrywide covariates which make infection trajectories of the countries behave differently in terms of the spread of the disease.

## Methods

### Richards growth curve models

Richards growth curve model [12], so-called the generalized logistic curve [13], is a growth curve model for population studies in situations where growth is not symmetrical about the point of inflection [14, 15]. The curve was widely used to describe various biological processes [16], but recently adapted in epidemiology for real-time prediction of outbreak of diseases; examples include SARS [17, 18], dengue fever [19, 20], pandemic influenza H1N1 [21], and COVID-19 outbreak [22].

There are variant reparamerized forms of the Richards curve in the literature [23–26], and we shall use the following form in this research

$$f(t; \theta_1, \theta_2, \theta_3, \xi) = \theta_1 \cdot [1 + \xi \cdot \exp\{-\theta_2 \cdot (t - \theta_3)\}]^{-1/\xi}, \tag{1}$$

where $\theta_1$, $\theta_2$, and $\theta_3$ are real numbers, and $\xi$ is a positive real number. The utility of the Richards curve (1) is its ability to describe a variety of growing processes, endowed with strong flexibility due to the shape parameter $\xi$ [24]: analytically, the Richards curve (1) (i) becomes the logistic growth curve [27] when $\xi = 1$, and (ii) converges to Gompertz growth curve [28] as the $\xi$ converges to zero from positive side of real numbers. (Gompertz curve is $g(t; \theta_1, \theta_2, \theta_3) = \theta_1 \cdot \exp[-\exp\{-\theta_2 \cdot (t - \theta_3)\}]$.) But it is also known that estimation of $\xi$ is a complicated problem [29], and we resort to a modern sampling scheme, elliptical slice sampler [30], to estimate the $\xi$. (See S3 Appendix for more detail).

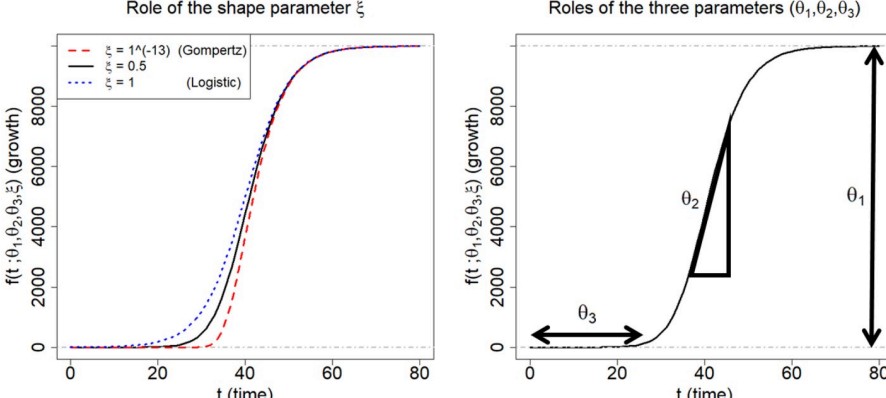

**Fig 3. Description of the Richards growth curve model.** The curve is obtained when $(\theta_1, \theta_2, \theta_3) = (10000, 0.2, 40)$. The left panel is obtained by changing the $\xi$ to be $1 \times 10^{-13}$, 0.5, and 1, respectively. The right panel describes the roles the three parameters in epidemiological modeling: $\theta_1$ represents final epidemic size; $\theta_2$ is an infection rate; and $\theta_3$ is a lag phase.

Fig 3 illustrates roles of the four parameters of the Richards curve (1). The curves on left panel is obtained when $(\theta_1, \theta_2, \theta_3) = (10000, 0.2, 40)$, while varying the $\xi$ to be $1 \times 10^{-13} (\approx 0)$, 0.5, and 1, respectively. The right panel pictorially describes the roles of $(\theta_1, \theta_2, \theta_3)$: $\theta_1$ represents the asymptote of the curve; $\theta_2$ is related to a growth rate; and $\theta_3$ sets the displacement along the x-axis. (For more technical detail for the parameters, refer to [24]).

In epidemiological modeling, the Richards curve (1) can be used as a parametric curve describing infection trajectories shown in the Fig 1. In this context, each of the parameters can be interpreted as follows: $\theta_1$ represents the final epidemic size (that is, the maximum cumulative number of infected cases across the times); $\theta_2$ represents infection rate; and $\theta_3$ represents a lag phase of the trajectory. (The shape parameter $\xi$ seems to have no clear epidemiological meaning [31]).

## Bayesian hierarchical Richards model

We propose a Bayesian hierarchical model based on the Richards curve (1), which is referred to as Bayesian hierarchical Richards model (BHRM), to accommodate the COVID-19 data $\{\mathbf{y}_i, \mathbf{x}_i\}_{i=1}^{N}$. (Although the model is based on the Richards curve, the idea can be generalized to any choice for growth curves.) Ultimately, a principal goal of the BHRM is to establish two functionalities:

(a). [Extrapolation] uncover a hidden pattern from the infection trajectory for each country $i$, that is, $\mathbf{y}_i = (y_{i1}, \cdots, y_{iT})^{\top}$, through the Richards growth curve $f(t; \theta_1, \theta_2, \theta_3, \xi)$ (1), and then extrapolate the curve.

(b). [Covariates analysis] identify important predictors among the $p$ predictors $\mathbf{x} = (x_1, \cdots, x_p)^{\top}$ that largely affect on the shape the curve $f(t; \theta_1, \theta_2, \theta_3, \xi)$ in terms of the three curve parameters.

A hierarchical formulation of the BHRM is given as follows. First, we introduce an additive independently identical Gaussian error to each observation $\{y_{it}\}_{i=1,t=1}^{N,T}$, leading to a likelihood part:

$$y_{it} = f(t; \theta_{1i}, \theta_{2i}, \theta_{3i}, \xi_i) + \epsilon_{it}, \quad \epsilon_{it} \sim \mathcal{N}(0, \sigma^2), \qquad (i = 1, \cdots, N, \ t = 1, \cdots, T), \qquad (2)$$

where $f(t;\theta_{1i}, \theta_{2i}, \theta_{3i}, \xi_i)$ is the Richards growth curve (1) which describes a growth pattern of infection trajectory for the $i$-th country. Because each of the curve parameters $(\theta_1, \theta_2, \theta_3)$ has its own epidemiological interpretations, we construct three separate linear regressions:

$$\theta_{li} = \alpha_l + \mathbf{x}_i^\top \boldsymbol{\beta}_l + \varepsilon_{li}, \quad \varepsilon_{li} \sim \mathcal{N}(0, \sigma_l^2), \qquad\qquad (i = 1, \cdots, N,\ l = 1, 2, 3), \qquad (3)$$

where $\boldsymbol{\beta}_l = (\beta_{l1}, \cdots, \beta_{lj}, \cdots, \beta_{lp})^\top$ is a $p$-dimensional coefficient vector corresponding to the $l$-th linear regression.

For the shape parameter $\xi$, we assume the standard log-normal prior:

$$\xi_i \sim \log \mathcal{N}(0, 1), \qquad (i = 1, \cdots, N). \qquad (4)$$

The motivation of choosing the log-normal prior (4) for the $\xi_i$ is that the prior puts effectively enough mass on the region $(0, 3)$ where most of the estimates for the $\xi_i$ $(i = 1, \cdots, N)$ concentrated on. Additionally, Gaussianity prior assumption makes it possible to employ the elliptical slice sampler [30] in sampling from the full conditional posterior distribution of the $\xi_i$.

To impose a continuous shrinkage effect [32] on each of the coefficient vectors, we adopt to use the horseshoe prior [33, 34]:

$$\beta_{lj}|\lambda_{lj}, \tau_{lj}, \sigma_l^2 \sim \mathcal{N}(0, \sigma_l^2 \tau_l^2 \lambda_{lj}^2), \quad \lambda_{lj}, \tau_{lj} \sim \mathcal{C}^+(0, 1), \quad (l = 1, 2, 3, j = 1, \cdots, p). \qquad (5)$$

Finally, improper priors [35] are used for the intercept terms and error variances terms in the model:

$$\alpha_l \sim \pi(\alpha) \propto 1, \quad \sigma^2 \quad , \sigma_l^2 \sim \pi(\sigma^2) \propto 1/\sigma^2, \qquad (l = 1, 2, 3). \qquad (6)$$

Generally speaking, modeling framework of the BHRM (2)–(6) is widely called the *nonlinear mixed effects model* or *hierarchical nonlinear model*, a standard framework for analysis of data in the form of continuous repeated measurements over time on each individual from a sample of individuals drawn from a population of interest [36]. See S3 Appendix for a posterior computation for the BHRM (2)–(6).

## Results

### Benefits from the information borrowing

We investigate the predictive performance of three Bayesian models based on the Richards growth curve. We start with the individual country-based model ($\mathcal{M}_1$) which has been widely used in the literature, reflecting the belief that individual countries are "unrelated." Next, we extend the previous model to a hierarchical model by utilizing the infection trajectories of all the 40 countries ($\mathcal{M}_2$). A limitation of $\mathcal{M}_2$ is that it lacks certain countrywide adjustments in estimating the trajectories. Next, we further upgrade this model by adding country-specific covariates in a hierarchical fashion ($\mathcal{M}_3$). (For technical description for the three models, see S2 Appendix) Eventually, borrowing information across the 40 countries takes place in these two hierarchical models, $\mathcal{M}_2$ and $\mathcal{M}_3$, but not in the individual country-based model $\mathcal{M}_1$.

For evaluation criteria, we calculate the mean squared error (MSE) [37] associated with the extrapolated infection trajectory for each of the 40 countries. Training and test data are designated as follows: given that $\mathbf{y}_k = (y_{k,1}, \cdots, y_{k,T})^\top$ is an infection trajectory of the $k$-th country spanning for $T$ days since January 22nd, and $d$ is the chosen test-day, then (i) the training data is set by the trajectory spanning for $T - d$ days since January 22nd (that is, $(y_{k,1}, \cdots, y_{k,T-d})$), and (ii) the test data is set by the $d$ recent observations (that is, $(y_{k,T-d+1}, \cdots, y_{k,T})$).

For the two hierarchical models $\mathcal{M}_2$ and $\mathcal{M}_3$, the MSE is averaged over the 40 countries:

$$\mathrm{MSE}_d = \frac{1}{40 \cdot d} \sum_{k=1}^{40} \sum_{r=T-d+1}^{T} (y_{k,r} - y_{k,r}^*)^2,$$

where $y_{k,r}$ is the actual value for the cumulative confirmed cases of the $k$-th country at the $r$-th time point, and $y_{k,r}^*$ is the forecast value: more concretely, $y_{k,r}^*$ is the posterior predictive mean given the information from 40 countries. For the non-hierarchical model $\mathcal{M}_1$, the $y_{k,r}^*$ in the $\mathrm{MSE}_d$ is acquired by using the data from the $k$-th country.

For each of the test-days ($d = 2, 4, \cdots, 28$), we report the $\mathrm{MSE}_d$'s from 50 replicates by showing the box plot. (The box plot [38] displays the distribution of $\mathrm{MSE}_d$ fore each d and model. The interquartile range (IQR) is represented as a box, which is from 25th percentile (Q1) to 75th percentile (Q3). A horizontal line in the box corresponds to the median. The maximun and minimun are set as $Q3 + 1.5 * IQR$ and $Q1 - 1.5 * IQR$, respectively. Any point larger than the maximun or smaller than the minumun is regarded as an outlier and drew as a black point in the plot.) The results are shown in Fig 4. From the panel, we see that (1) the predictive performances of two hierarchical models, $\mathcal{M}_2$ and $\mathcal{M}_3$, are better than that of $\mathcal{M}_1$ across the test-days; (2) the differences in the predictive performance between the non-hierarchical ($\mathcal{M}_1$) and the hierarchical models ($\mathcal{M}_2$ and $\mathcal{M}_3$) tend to get larger as the test-days increase; and (3) the predictive performances of two hierarchical models ($\mathcal{M}_2$ and $\mathcal{M}_3$) are similar across the test-days. Based on the outcomes, we shall conclude that information borrowing has improved the predictive accuracy in terms of MSE. A similar result where information borrowing is a benefit in improving predictive accuracy is found in the *Clemente problem* from [39] where the James-Stein estimator [40] better predicts then an individual hitter-based estimator in terms of the

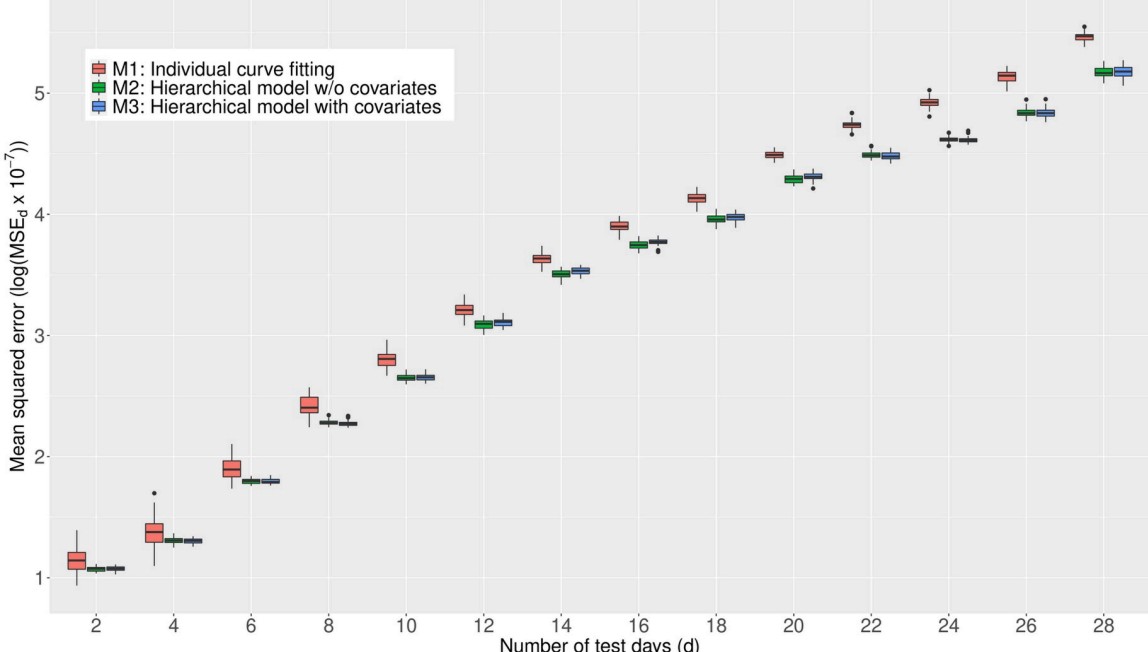

**Fig 4. Comparison of the MSE obtained by the three models, $\mathcal{M}_1$, $\mathcal{M}_2$, and $\mathcal{M}_3$, averaged over the 40 countries.** A smaller value for the MSE indicates a better predictive performance.

total squared prediction error. In what follows, we present all the results in the consequent subsections based on the model $\mathcal{M}_3$.

## COVID-19 travel recommendations by country

Centers for Disease Control and Prevention (CDC) categorizes countries into three levels by assessing the risk of COVID-19 transmission, used in travel recommendations by country (Visit www.cdc.gov/-): Level 1, Level 2, and Level 3 indicate the Watch Level (Practice Usual Precautions), Alert Level (Practice Enhanced Precautions), and Warning Level (Avoid Nonessential Travel), respectively.

We categorize the 40 countries into the three levels according to their posterior means for the final epidemic size (that is, $\theta_1$ of the Richards curve (1)). Grouping criteria are as follows: (1) Level 1 (estimated total number is no more than 10,000 cases); (2) Level 2 (estimated total number is between 10,000 and 100,000 cases); and (3) Level 3 (estimated total number is more than 100,000 cases).

Fig 5 displays results of posterior inference for the $\theta_1$ by country, based on the model $\mathcal{M}_3$. Countries on the $y$-axis are ordered from the severest country (US) to the least severe country (Malaysia) in the magnitude of the posterior means. The red horizontal bars on the panel represent the 95% credible intervals describing the uncertainty regarding the estimated final epidemic size $\theta_1$. (Technically, lower and upper bounds of each of the intervals are obtained by taking the 2.5-th and 97.5-th percentiles from posterior samples of $\theta_1$ for the corresponding country, respectively.) Based on the results, there are 14 countries categorized as Level 3 (US, Russia, Brazil, Pakistan, UK, Spain, Italy, India, France, Germany, Peru, Iran, Chile, and Canada). There are 21 countries categorized as Level 2 (from Saudi Arabia to South Korea), and 5 countries categorized as Level 1 (from Czechia to Malaysia).

## Extrapolated infection trajectories and flat time points

Fig 6 displays the extrapolated infection trajectory (posterior mean for the Richards curve (1)) for the US. The posterior mean of the final epidemic size is 1,760,569 cases. The scenario that 'millions' of Americans could be infected was also warned by a leading expert in infectious

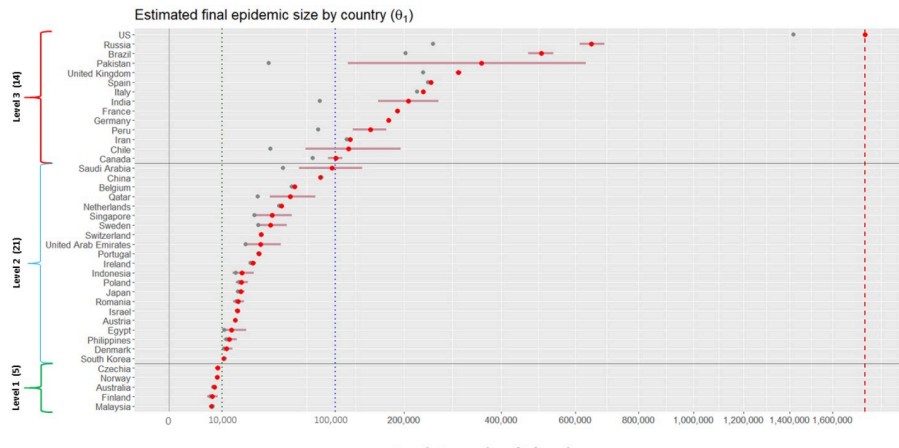

**Fig 5. Estimation results for the final epidemic size for 40 countries.** Grey dots (•) represent the cumulative numbers of infected cases for 40 countries on May 14th; red dots (•) and horizontal bars (—) represent the posterior means and 95% credible intervals for the $\theta_1$ of the 40 countries. Vertical red dotted line indicates the 1, 760, 569 cases, the posterior mean for the US.

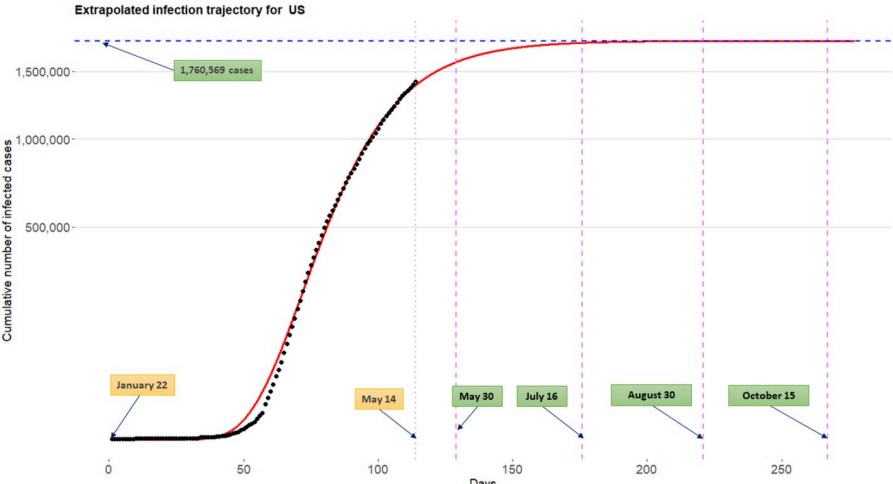

**Fig 6. Extrapolated infection trajectory for the US based on the model $\mathcal{M}_3$.** Posterior mean of the maximum number of cumulative infected cases is is 1,760,569 cases. Posterior means for the flat time points are $t_{\text{flat},\gamma=0.9}$ = May 30th, $t_{\text{flat},\gamma=0.99}$ = July 16th, $t_{\text{flat},\gamma=0.999}$ = August 30th, and $t_{\text{flat},\gamma=0.9999}$ = October 15th.

diseases (Visit a related news article www.bbc.com/-). It is known that prediction of an epidemic trend from limited data during early stages of the epidemic is often futile and misleading [17]. Nevertheless, estimation of a possible severity havocked by the COVID-19 outbreak is an important task when considering the seriousness of the current pandemic situation.

A crucial question is when this trajectory gets flattened. To that end, we approximate a time point where at an infection trajectory levels off its value, showing a flattening pattern after the time point. The following is the definition of the *flat time point* which we use in this paper:

**Definition 0.1** *Consider the Richards curve $f(t;\theta_1, \theta_2, \theta_3, \xi)$ (1). Given a progression constant $\gamma$(%) with $0 < \gamma < 1$, the flat time point $t_{\text{flat},\gamma}$ is defined as the solution of the equation*

$$\gamma = \frac{f(t; \theta_1, \theta_2, \theta_3, \xi)}{\theta_1} \ (\%). \tag{7}$$

By using elementary calculus, we can obtain the solution of the Eq (7):

$$t_{\text{flat},\gamma} = \theta_3 - \frac{1}{\theta_2} \cdot \log\left[\frac{1}{\xi} \cdot \left\{\left(\frac{1}{\gamma}\right)^{\xi} - 1\right\}\right], \quad 0 < \gamma < 1. \tag{8}$$

Technically, the flat time point $t_{\text{flat},\gamma}$ (7) is interpreted as follow. Given a progression constant $\gamma$% (set by epidemiologist), the flat time point $t_{\text{flat},\gamma}$ is the time point whereat only $(1 - \gamma)$ $\theta_1$ cases can maximally take place to reach the final epidemic size $\theta_1$ following the time point $t_{\text{flat},\gamma}$. Here, the progression constant $\gamma$(%) is a value indicating a development of the pandemic: a higher value of $\gamma$ implies a later stage of the pandemic where at infection trajectories begin or tend to reach plateau. Fig 7 depicts an exemplary infection trajectory based on the Richards curve (1) where the parameters were chosen by $(\theta_1, \theta_2, \theta_3, \xi)$ = (10000, 0.2, 40, 0.5), while the progression constant is set by $\gamma$ = 0.9, leading to flat time point $t_{\text{flat},\gamma}$ to be approximately 51.

Standard choices for $\gamma$ shall be 0.9, 0.99, 0.999, etc, because infection trajectories typically begins to display its flattening phase when $\gamma$ is equal to or greater than 0.9. Choice of value for $\gamma$ depends on the particular situation of a country considered: for example, for China which already shows flattened trajectory (refer to Fig 1), $\gamma$ = 0.999 can be safely used, but for US one

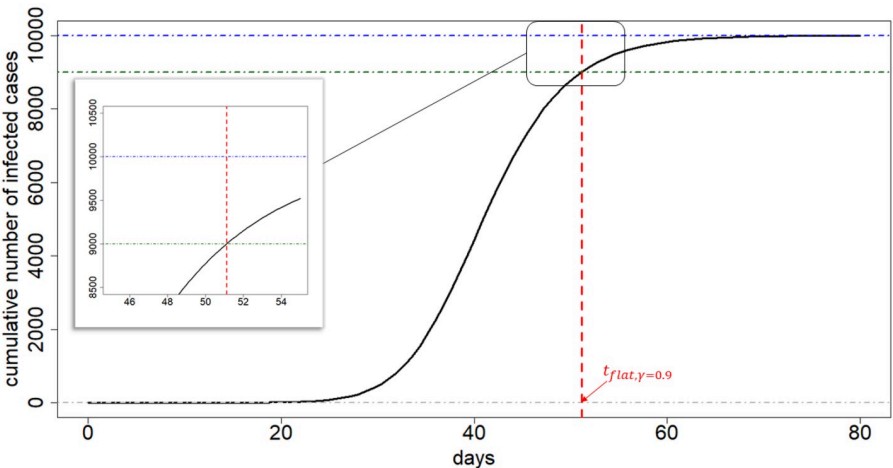

**Fig 7. Illustration of flat time point.** The exemplary infection trajectory is obtained by the Richards curve when $(\theta_1, \theta_2, \theta_3, \xi) = (10000, 0.2, 40, 0.5)$. A flat time point $t_{\text{flat},\gamma}$ is approximately 51 (vertical red dashed line). The vertical difference between the $\theta_1$ and the function value of Richards curve evaluated at $t_{\text{flat},\gamma}$ is $\gamma = 0.9$.

may use each of the $\gamma = 0.9, 0.99, 0.999,$ and $0.9999$ to further investigate evolvement of flattening phase over time.

For the US, the posterior means of the flat time points $t_{\text{flat},\gamma}$ are May 30th, July 16th, August 30th, and October 15th when corresponding $\gamma$'s are chosen by 0.9, 0.99, 0.999, and 0.9999, respectively. It is important to emphasize that the extrapolated infection trajectory is *real-time prediction* of COVID-19 outbreaks [31, 41] based on observations tracked until May 14th. Certainly, incorporation of new information such as compliance with social distancing or advances in medical and biological sciences for this disease will change the inference outcomes.

Fig 8 shows the extrapolated infection trajectories for Russia, UK, and Brazil. Posterior means of the final epidemic size are as follows: (1) for the Russia, 648,190 cases; (2) for the UK, 303,715 cases; and (3) for the Brazil, 503,271 cases. Flat time points are estimated by: (1) for the Russia, $t_{\text{flat},\gamma=0.9}$ = June 27th, $t_{\text{flat},\gamma=0.99}$ = August 11th, $t_{\text{flat},\gamma=0.999}$ = September 24th, and $t_{\text{flat},\gamma=0.9999}$ = November 6th; (2) for the UK, $t_{\text{flat},\gamma=0.9}$ = June 2nd, $t_{\text{flat},\gamma=0.99}$ = July 19th, $t_{\text{flat},\gamma=0.999}$ = September 3rd, and $t_{\text{flat},\gamma=0.9999}$ = October 19th; and (3) for the Brazil, $t_{\text{flat},\gamma=0.9}$ = June 16th, $t_{\text{flat},\gamma=0.99}$ = July 16th, $t_{\text{flat},\gamma=0.999}$ = August 15th, and $t_{\text{flat},\gamma=0.9999}$=September 13th. Results for other countries are included in the S4 Appendix.

## Global trend for the COVID-19 outbreak

Fig 9 displays the extrapolated infection trajectory for grand average over 40 countries obtained from the model $\mathcal{M}_3$. Technically, this curve is acquired by extrapolating the Richards curve by using the intercept terms in linear regressions (3). The grey dots on the panel are historical infection trajectories for 40 countries. Posterior means for the final epidemic size is 145,497 cases. Posterior means for the flat time points are $t_{\text{flat},\gamma=0.999}$=July 5th and $t_{\text{flat},\gamma=0.9999}$ = July 31st.

## Identifying risk factors for severe disease due to COVID-19

COVID-19 is a new disease and there is very limited information regarding risk factors for this severe disease. There is no vaccine aimed to prevent the transmission of the disease because there is no specific antiviral agent is available. It is very important to find risk factors relevant to the disease. Reliable and early risk assessment of a developing infectious disease outbreak

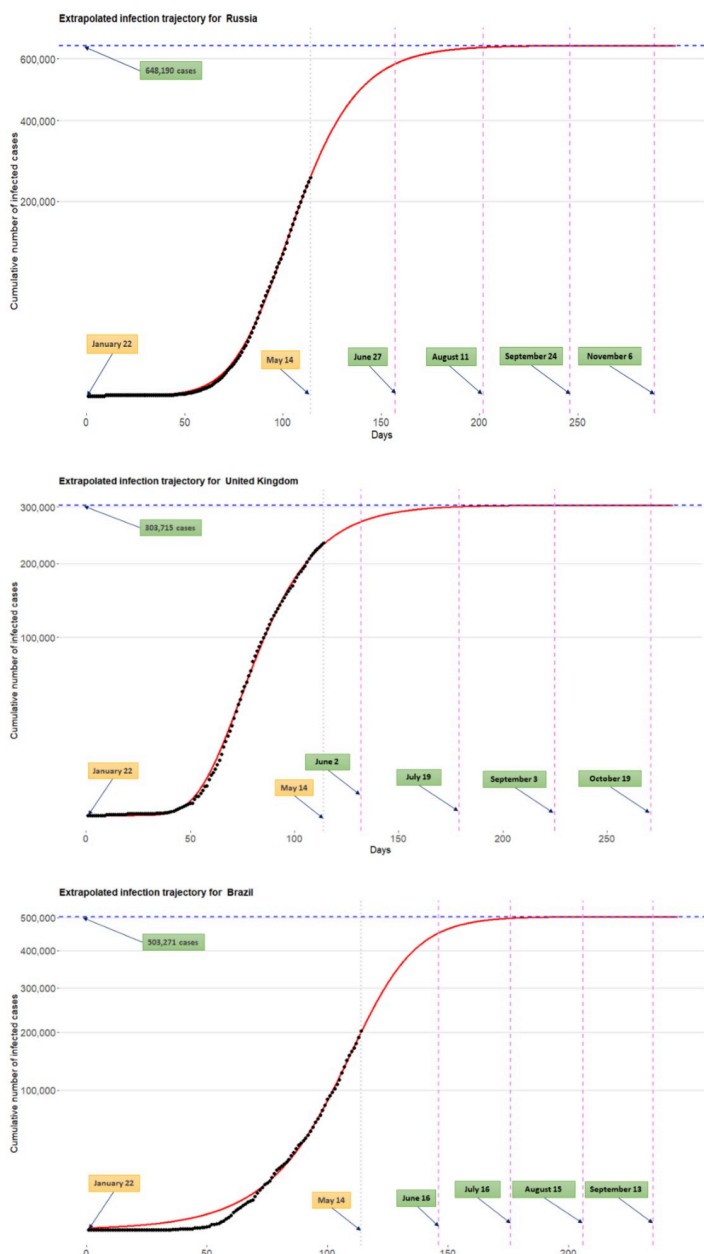

**Fig 8. Extrapolated infection trajectory for the Russia (top), UK (middle), and Brazil (bottom).** Flat time points are estimated by: (1) for the Russia, $t_{\text{flat},\gamma=0.9}$ = June 27th, $t_{\text{flat},\gamma=0.99}$ = August 11th, $t_{\text{flat},\gamma=0.999}$ = September 24th, and $t_{\text{flat},\gamma=0.9999}$ = November 6th; (2) for the UK, $t_{\text{flat},\gamma=0.9}$ = June 2nd, $t_{\text{flat},\gamma=0.99}$ = July 19th, $t_{\text{flat},\gamma=0.999}$ = September 3rd, and $t_{\text{flat},\gamma=0.9999}$ = October 19th; and (3) for the Brazil, $t_{\text{flat},\gamma=0.9}$ = June 16th, $t_{\text{flat},\gamma=0.99}$ = July 16th, $t_{\text{flat},\gamma=0.999}$ = August 15th, and $t_{\text{flat},\gamma=0.9999}$ = September 13th.

allow policymakers to make swift and well-informed decisions that would be needed to ensure epidemic control.

CDC described High-Risk Conditions based on currently available information and clinical expertise (For more detail, visit www.cdc.gov/-): those at higher risk for infection, severe illness, and poorer outcomes from COVID-19 include.

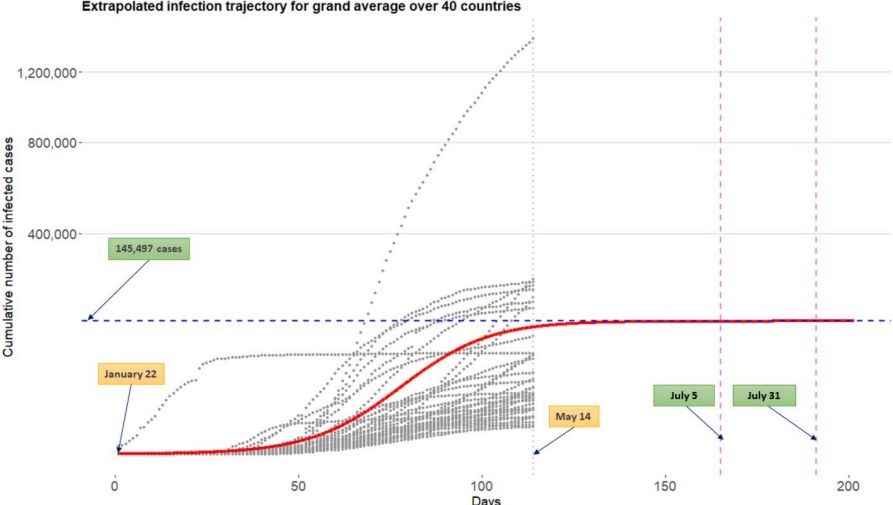

**Fig 9. Extrapolated infection trajectory for grand average over 40 countries obtained from the model $\mathcal{M}_3$.** Grey dots are historical infection trajectories for 40 countries spanning from January 22nd to May 14th. Posterior means for the flat time points are $t_{\text{flat},\gamma=0.999}$ = July 5th and $t_{\text{flat},\gamma=0.9999}$ = July 31st.

- People 65 years and older;

- People who live in a nursing home or long-term care facility;

- People with chronic lung disease or moderate to severe asthma;

- People who are immunocompromised, possibly caused by cancer treatment, smoking, bone marrow or organ transplantation, immune deficiencies, poorly controlled HIV or AIDS, and prolonged use of corticosteroids and other immune weakening medications;

- People with severe obesity (body mass index of 40 or higher);

- People with diabetes;

- People with chronic kidney disease undergoing dialysis;

- People with liver disease.

The model $\mathcal{M}_3$ involves three separated linear regressions indexed by $l$ = 1, 2, and 3, whose response and coefficient vector are denoted by $\theta_l$ and $\boldsymbol{\beta}_l$, respectively ($l$ = 1, 2, 3). (See the Eq (3)) The sparse horseshoe prior [33, 34] is imposed for each of the coefficient vectors, which makes the model equipped with covariates analysis. That way, we can identify important predictors explaining the heterogeneity of shapes existing in infection trajectories across 40 countries. Because each of the responses has its own epidemiological interpretation (final epidemic size ($\theta_1$), infection rate ($\theta_2$), and lag phase ($\theta_3$)), the joint variable selection techniques employed by the sparse horseshoe prior can be further used in finding possible risk factors for severe disease due to COVID-19 among the 45 predictor considered in this research.

Table 1 summarizes 10 significant predictors among the 45 predictors, explaining each of the responses $\theta_l$ ($l$ = 1, 2, 3). Contents of the table are listed with the form "covariate name (estimate of coefficient)". Technically, the estimate inside of the parenthesis is the posterior mean for the coefficient. By following convention in variable selection schemes as done by several authors [42, 43], we standardized the design matrix and then make a posterior inference:

**Table 1. Important predictors explaining $\theta_l$, $l$ = 1, 2, 3.**

| Final epidemic size ($\theta_1$) | Infection rate ($\theta_2$) | Lag phase ($\theta_3$) |
|---|---|---|
| Insuf_phy_act(+19895) | Points_of_Entry(−0.0107) | Dis_to_China(+16.14) |
| Testing_num(+19256) | Alcohol_consumers_total(+0.0106) | Alcohol_cons_rec(−5.24) |
| Testing_popu(−16596) | Alcohol_cons_rec(+0.0071) | Median_age(−4.70) |
| Overweight(+16173) | Air_pollution(+0.0055) | Alcohol_consumers_total(−4.33) |
| MCV1_immun(−15447) | Life_expect_total_60(+0.0054) | Testing_num(−4.16) |
| Testing_confirm(+10689) | Popu_density(+0.0050) | Life_expect_total_60(−1.93) |
| Pol3_immun(−8388) | Laboratory(−0.0046) | Cigarette_smoke(−1.85) |
| Hib3_immun(−7478) | Heavy_drinking_total(+0.0044) | Tuberculosis_case(−1.80) |
| Tempe_avg(−4573) | Cigarette_smoke(+, 0.0026) | Total_over_65(−1.35) |
| Alcohol_cons_rec(+4205) | Tobacco_smoke(+0.0025) | Tobacco_smoke(−1.29) |

The table shows 10 interesting covariates for each parameter. They are listed with "covariate name (estimate of coefficient)" where the estimate is the posterior mean for the corresponding coefficient. See S1 Appendix for a detailed explanation for the listed covariates.

therefore, an estimated coefficient may not indicate a change of value for the response $\theta_l$ ($l$ = 1, 2, 3) with respect to a unit increment of value for covariate. Rather, the estimate can be thought of as a measure representing the sensitiveness of a change in response with respect to a change in the covariate.

The followings are general guidelines on how covariates on the Table 1 can be interpreted in the current context of pandemic.

- The parameter $\theta_1$ represents *final epidemic size*. A larger number of $\theta_1$ indicates that a country has (can have) more COVID-19 infected patients in the country. A covariate with a positive estimate (or negative estimate) is a factor associated with an increase (or decrease) of the total infected cases. A covariate endowed with a larger magnitude (that is, absolute value) for the estimate makes a greater influence on $\theta_1$.

- The parameter $\theta_2$ represents *infection rate*. A larger number of $\theta_2$ implies a faster spread of the virus around the country. A covariate with a positive estimate (or negative estimate) is a factor associated with a rapid (or slow) spread of the virus. A covariate endowed with a larger magnitude for the estimate makes a greater influence on $\theta_2$.

- The parameter $\theta_3$ represents *lag phase* of the infection trajectory. The larger the value of $\theta_3$ the later the trajectory begins to accumulate infected cases, leading to a later onset of the accumulation. A covariate with a positive estimate (or negative estimate) is a factor associated with delaying (or bring forward) the onset of the accumulation. A covariate endowed with a larger magnitude for the estimate makes a greater influence on $\theta_3$.

Now, based on the aforementioned guideline, we shall interpret the Table 1 in detail. (The reasoning reflects our subjectivity, and disease expert should decipher precisely).

As for the parameter $\theta_1$, insufficient physical activity has been selected as one of the important risk factors which may increase the final epidemic size of a country. Additionally, intense immunization coverage on measles, Polio, and Haemophilus Influenzae type B can reduce the final epidemic size. Poor general health status of a population [44] such as overweight and alcohol addiction can increase the epidemic size. (visit related news article www.cidrap.umn.edu/-.) Certain testing information is also associated with the epidemic size, which can be further researched in retrospective studies in swift policymaking for a future pandemic. (See a WHO report for the relationship between climate change and infectious diseases www.who.int/-).

Turning to the parameter $\theta_2$, a rigorous fulfillment of general obligations at point of entry is chosen as one of the significant predictors in reducing the infection rate. Additionally, poor smoking and alcoholic behaviors of a country population are risk factors that may increase the infection rate. Demographically, it has been found that densely populated countries or countries where life expectancy is relatively high are more venerable to the rapid disease transmission among people. Among national environmental status, poor air condition which may negatively influence people's respiratory system is found to be a risk factor increasing the infection rate.

Finally, moving to the parameter $\theta_3$, a geological distance of a certain country from China is an important covariate delaying the onset of the infected cases. The lag of onset is also graphically observed from the Fig 1: time point whereat South Korea begins to accumulate the infected cases is relatively earlier than those of the US, UK, etc. Similar to $\theta_2$, heavier alcohol drinking and tobacco use may result in an earlier onset of the accumulation of the infected patients, thereby bringing forward the infection trajectory. Having larger numbers of median age and elderly people of a population can shorten the lag phase. Finally, conducting frequent testing for the COVID-19 helps detect infected patients, followed by the earlier accumulation for the confirmed cases.

## Discussions

In general, there are three major categories of infectious disease prediction models: (i) differential equation models, (ii) time series models, and (iii) the statistical models. The differential equation models describe the dynamic behavior of the disease through differential equations allowing the laws of transmission within the population. The popular models include the SI, SIS, SIR, and SEIR models [45–47]. These models are based on assumptions related to S (susceptible), E (exposed), I (infected), and R (remove) categories of the population. Time series based prediction models such as ARIMA, Grey Model, Markov Chain models have been used to describe dependence structure over of the disease spread over time [48–52]. On the other hand, statistical models, so-called phenomenological models, which follow certain laws of epidemiology [53, 54] are widely used in real-time forecasting for infection trajectory or size of epidemics in early stages of pandemic [18, 41, 55]. Statistical models can be easily extended to the framework of hierarchical models (multilevel models [56]) to analyze data within a nested hierarchy, eventually harnessing the data integration [57–60]. In this paper, we proposed a Bayesian hierarchical model, BHRM (2)–(6), so that data integration and uncertainty analysis [61] are possible in a unified way.

BHRM is a Bayesian version of a two-stage non-linear mixed effect model [36] where the first and second stages are related to the curve-fitting based on a certain parametric curve (we used Richards curve (1)) and covariates analysis, respectively. In such a non-linear modeling framework, one of the challenges is an accommodation of individual-specific covariates which change over the course of observations. (See page 149 in [36] and [62] for more detailed discussion on this issue.) However, in infectious disease modeling for COVID-19 spread curves, there are important time-varying covariates that can be further considered in a possible model: examples include daily number of COVID-19 tests conducted, daily social distancing scores, etc. These time-varying covariates may be used in data level (likelihood level) endowed with center-adjusted inference for the curve-fitting [63].

It is important to point out that the real-time forecast during early stages of the pandemic may result in premature inference outcomes [17], but it should not demoralize predictive analysis as the entire human race is currently threatened by unprecedented crisis due to COVID-19 pandemic. To improve the predictive accuracy, data integration from multiple countries was a key notion, which is closely related to borrowing information. The motivation of using

the borrowing information is to make use of *indirect evidence* [39] to enhance the predictive performance: for example, to extrapolate the infection trajectory for the US, the information not only from the US (*direct evidence*) but also from other countries (*indirect evidence*) are utilized to better predict the trajectory for the US. Further, to render the information borrowing endowed with uncertainty quantification, Bayesian argument is inevitable, inducing sensible inferences and decisions for users [64].

## Conclusion

It is important to emphasize that, while medical and biological sciences are on the front lines of beating back COVID-19, the true victory relies on advance and coalition of almost every academic field. However, information about COVID-19 is limited: there are currently no vaccines or other therapeutics approved by the US Food and Drug Administration to prevent or treat COVID-19 (on April 13, 2020). Although numerous research works are progressed by different academic fields, the information about COVID-19 is scattered around different disciplines, which truly requires interdisciplinary research to hold off the spread of the disease.

In this paper, we proposed the BHRM (2)–(6) based on the Richards growth curve (1) [12]. In summary, the novelties of our method are as follows: we (i) used a flexible hierarchical growth curve model to global COVID-19 data, (ii) integrated information from 40 countries for estimation and prediction purposes, and (iii) performed covariate analysis to find important reasons to explain the heterogeneity in the country-wise infection trajectories across 40 countries.

The results demonstrated the superiority of our approach compared to an existing individual country-based model. Our research outcomes can be thought even more insightful given that we have not employed information about disease-specific covariates. That being said, using more detailed information such as social mixing data, precise hospital records, or patient-specific information will further improve the performance of our model. Moreover, integration of epidemiological models with these statistical models will be our future topic of research.

## Supporting information

**S1 Appendix. Research data.**
(PDF)

**S2 Appendix. Technical expressions for the three models $\mathcal{M}_1$, $\mathcal{M}_2$, and $\mathcal{M}_3$.**
(PDF)

**S3 Appendix. Posterior computation.**
(PDF)

**S4 Appendix. Infection trajectories for the top 20 countries.**
(PDF)

**S1 File. Minimal underlying data set and relevant execution codes.**
(ZIP)

## Author Contributions

**Conceptualization:** Bani Mallick.

**Data curation:** Bowen Lei.

**Formal analysis:** Se Yoon Lee.

**Investigation:** Bowen Lei.

**Methodology:** Se Yoon Lee, Bani Mallick.

**Project administration:** Bani Mallick.

**Resources:** Bowen Lei.

**Software:** Bowen Lei.

**Supervision:** Bani Mallick.

**Visualization:** Se Yoon Lee, Bowen Lei.

**Writing – original draft:** Se Yoon Lee.

**Writing – review & editing:** Se Yoon Lee, Bani Mallick.

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
