## [Decision Letter · Decision Letter 0]

22 Jun 2020

PONE-D-20-14650

Estimation of COVID-19 spread curves integrating global data and borrowing information

PLOS ONE

Dear Dr. Lee,

Thank you very much for submitting your manuscript "Estimation of COVID-19 spread curves integrating global data and borrowing information" (#PONE-D-20-14650) for review by PLOS ONE. As with all papers submitted to the journal, your manuscript was fully evaluated by academic editor (myself) and by independent peer reviewers. The reviewers appreciated the attention to an important health topic, but they raised substantial concerns about the paper that must be addressed before this manuscript can be accurately assessed for meeting the PLOS ONE criteria. Therefore, if you feel these issues can be adequately addressed, we invite you to submit a revised version of the manuscript that addresses the points raised during the review process. We can’t, of course, promise publication at that time.

We look forward to receiving your revised manuscript.

Kind regards,

Abdallah M. Samy, PhD

Academic Editor

PLOS ONE

**Journal Requirements:**

'The funders had no role in study design, data collection and analysis, decision to publish, or preparation of the manuscript.'

Please amend your Financial disclosure statement to declare sources of funding, or state that the authors received no specific funding.

5. We note you have included tables to which you do not refer in the text of your manuscript. Please ensure that you refer to Tables 3, 4, 5, 6, 7, 8 and 9 in your text; if accepted, production will need this reference to link the reader to each Table.

6. Please include a separate caption for each figure in your manuscript.

7. In your Data Availability statement, you have not specified where the minimal data set underlying the results described in your manuscript can be found. PLOS defines a study's minimal data set as the underlying data used to reach the conclusions drawn in the manuscript and any additional data required to replicate the reported study findings in their entirety. All PLOS journals require that the minimal data set be made fully available. For more information about our data policy, please see http://journals.plos.org/plosone/s/data-availability.

**Reviewers' comments:**

Reviewer's Responses to Questions

**Comments to the Author**

1. Is the manuscript technically sound, and do the data support the conclusions?

Reviewer #1: Yes

2. Has the statistical analysis been performed appropriately and rigorously?

Reviewer #1: Yes

3. Have the authors made all data underlying the findings in their manuscript fully available?

Reviewer #1: Yes

4. Is the manuscript presented in an intelligible fashion and written in standard English?

Reviewer #1: Yes

5. Review Comments to the Author

Reviewer #1: Overview:

The manuscript presents a statistical hierarchical model based on the Richards growth curve aiming to forecast the spread of SARS-Cov-2 in 40 countries. The work explores three different parametrizations of the model proposed and compare their results. Epidemic modelling is an important subject and due to the COVID-19 pandemic we are living, this is even more significant as robust and accurate predictions are essential to support health policies. The manuscript is well written, clear and, easy to follow. Nonetheless, I have some questions and comments.

Comments/suggestions:

The remaining text is divided according to the manuscript sections and follows as close as possible the flow of the paper.

Results

To assess the models the authors used the MSE (mean square error) which was computed for a different number of the last observations. The authors computed the MSE for the last 5, 7, 8, 9, and 10 days of the period observed and, also for 20, 22, 24, 26, and 28 days. The training set was composed of the first values of the time series, and so, the MSE was calculated in a different set, the test set. This approach allows gauging the prediction capacity of the models, however, no explanation is provided regarding the reasons to select these groups of days. It seems that a more natural choice would be using just one interval of testing days spanning from 5 to 28 days, for example. Thus, the values could then be plotted in one single chart enabling to perceive the impact on the MSE of using more data in the training set. With two charts (figure 4), as the authors have presented, it is not possible to fully understand if the relationship between the MSE and the size of the training set is linear or non-linear. Another point is the use of the median, which although understandable because it is more robust than the average, no explanation is given for its use. In my opinion, besides the median, confidence bands could have been used, which would allow a more comprehensive comparison between the methods. Perhaps, M2 and M3 methods would be statistically equivalent considering the confidence bands, which in turn would lead to slightly different conclusions concerning these methods.

In figure 5, the confidence intervals of parameter theta 1 (final epidemic size) are presented however no details about the computation of the confidence intervals are provided. Please, describe how the confidence intervals were obtained.

The time to reach the final plateau of the epidemic is given by considering the final E (epsilon) cases before the epidemic size. This is a clever approach, and even better when considering different values for epsilon. However, this approach does not take into account the epidemic size. Did the authors contemplate different forms, for example, consider the time to reach a percentage of the epidemic size?

Table 1, shows the sign regarding different factors influencing the parameters theta 1, 2, and 3. For example, insufficient physical activity (positive sign) increases the final epidemic size, whereas temperature decreases it. This information is important but without quantification is very poor. In my opinion, Table 1 should present the average values of the coefficients of regression and, their interpretation should be done considering those values. It is completely different if the final epidemic size, given by theta 1, decreases in average 100 or decreases 10,000 cases per increase of one Celsius degree. The discussion following table 1, is, therefore, highly imprecise and can be misleading. Furthermore, the values for these covariates (shown in Appendix A) used for each country refer to distinct years, from 2013 to 2018. The authors should include a note concerning this aspect addressing the possible bias caused by these values.

Discussion

In my opinion, the discussion lacks two main aspects that are somehow related. On one hand, the discussion does not address any limitation of the work, and on the other, does not criticise the predictions presented regarding the possibility of error. Any model attempting to forecast the future is doomed to fail. The question is about how much and how can it be corrected. If we compare the current values and the ones concerning the epidemic size (e.g. figures 8 and 9) it is evident that the predictions have failed. I do not consider this a failure of the work. However, the discussion should mention this possibility as well as to address why this is likely to happen and how it can be mitigated.

Materials and methods

The selection of the 40 countries is unclear. Please, clarify what was the selection criterion.

Figures

In figure 4 the units in the y-axis are different, but if they represent the same entity shouldn't they be equal?

6. PLOS authors have the option to publish the peer review history of their article (what does this mean?). If published, this will include your full peer review and any attached files.

Reviewer #1: Yes: Francisco Caramelo

---

## [Author Response · Author response to Decision Letter 0]

7 Jul 2020

We have attached the file name "Response to reviewers" in word document. Please see our responses from the file.

---

## [Editor Report · Decision Letter 1]

16 Jul 2020

Estimation of COVID-19 spread curves integrating global data and borrowing information

PONE-D-20-14650R1

Dear Dr. Lee,

We’re pleased to inform you that your manuscript has been judged scientifically suitable for publication and will be formally accepted for publication once it meets all outstanding technical requirements.

Kind regards,

Abdallah M. Samy, PhD

Academic Editor

PLOS ONE

---

## [Editor Report · Acceptance letter]

21 Jul 2020

PONE-D-20-14650R1 

Estimation of COVID-19 spread curves integrating global data and borrowing information 

Dear Dr. Lee:

I'm pleased to inform you that your manuscript has been deemed suitable for publication in PLOS ONE. Congratulations! Your manuscript is now with our production department. 

Kind regards, 

on behalf of

Dr. Abdallah M. Samy 

Academic Editor

PLOS ONE